# PCBs, PCNs, and PCDD/Fs in Soil around an Industrial Park in Northwest China: Levels, Source Apportionment, and Human Health Risk

**DOI:** 10.3390/ijerph20043478

**Published:** 2023-02-16

**Authors:** Tianwei Li, Jicheng Hu, Chenyang Xu, Jun Jin

**Affiliations:** 1College of Life and Environmental Science, Minzu University of China, Beijing 100081, China; 2Key Laboratory of Ecology and Environment in Minority Areas, Minzu University of China, National Ethnic Affairs Commission, Beijing 100081, China; 3Beijing Engineering Research Center of Food Environment and Public Health, Minzu University of China, Beijing 100081, China

**Keywords:** polychlorinated biphenyls, polychlorinated naphthalenes, dioxins, soil contamination, source apportionment, industrial park

## Abstract

The concentrations of polychlorinated biphenyls (PCBs), polychlorinated naphthalenes (PCNs), and polychlorinated dibenzo-*p*-dioxins and dibenzofurans (PCDD/Fs) were determined in soil samples collected around an industrial park in Northwest China, to investigate the potential impacts of park emissions on the surrounding environment. The total concentration ranges of PCBs, PCNs, and PCDD/Fs in the soil samples were in 13.2–1240, 141–832, and 3.60–156 pg/g, respectively. The spatial distribution and congener patterns of PCBs, PCNs, and PCCD/Fs indicated that there might be multiple contamination sources in the study area, so source apportionments of PCBs, PCNs, and PCCD/Fs were performed by a positive matrix factorization model based on the concentrations of all target congeners together. The results revealed that these highly chlorinated congeners (CB-209, CN-75, and OCDF) might be derived from phthalocyanine pigments, the legacy of Halowax 1051 and 2,4-D products, which together contributed nearly half of the total concentration of target compounds (44.5%). In addition to highly chlorinated congeners, the local industrial thermal processes were mainly responsible for the contamination of PCBs, PCNs, and PCDD/Fs in the surrounding soil. The total carcinogenic risk of PCBs, PCNs, and PCDD/Fs in a few soil samples (0.22 × 10^−6^, 0.32 × 10^−6^, and 0.40 × 10^−6^) approached the threshold of potential carcinogenic risk (1.0 × 10^−6^). Since these pollutants can continuously accumulate in the soil, the contamination of PCBs, PCNs, and PCDD/Fs in surrounding soil deserves continuous attention.

## 1. Introduction

Polychlorinated biphenyls (PCBs), polychlorinated naphthalenes (PCNs), and polychlorinated dibenzo-*p*-dioxins and dibenzofurans (PCDD/Fs) are listed in the Stockholm Convention as persistent organic pollutants (POPs) because they are resistant to degradation, highly toxic, can be enriched through the food chain, and can globally migrate via various pathways [1,2]. Apart from scientific research, humans have never intentionally produced PCDD/Fs [3]. Instead, they are typically generated as impurities in various chemical products, such as chlorophenols and chlorophenoxy acid herbicides [4,5]. In contrast, PCNs and PCBs were mass produced and widely used as heat carriers and insulating oils (e.g., in capacitors and transformers) during the last century because they have good chemical stability, insulating properties, and heat resistance. Meanwhile, high concentrations of PCB congeners, some of which have not been detected in or are present in very low concentrations in commercial PCB formulations, such as PCB 11, have been detected in commercial organic pigments [6,7]. Furthermore, PCBs, PCNs, and PCDD/Fs can also be unintentionally formed in industrial thermal processes, such as metal smelting, waste incineration, and cement production [8,9,10]. Some PCN congeners, which are formed in thermal processes but present at very low concentrations in commercial PCN formulations, are termed combustion-related congeners. Since these compounds are highly toxic and their sources are extremely complex, the contamination characteristics and source apportionment of PCBs, PCNs, and PCDD/Fs in the environment have attracted much attention in past decades.

To intensify industrialization and optimize functional layouts, multiple industrial thermal processes are often concentrated in industrial parks. Commercial PCB and PCN formulations and products rich in these compounds are likely to have been used in these industrial parks. Therefore, environmental contamination with PCBs, PCNs, and PCDD/Fs is often complex around industrial areas. Because the congener patterns of PCBs, PCNs, and PCDD/Fs from different sources usually have unique profiles, they are often used to identify the sources of these pollutants in the environment. In addition, some congeners only from industrial products or specific sources, such as PCB 11 and PCN combustion-related congeners, can be used as key indicators of specific pollution sources in the environment. However, in most cases, because of the influence of multiple pollution sources, it is difficult to identify sources at a glance using only the PCB, PCN, and PCDD/F patterns of environmental samples. In some studies, the potential sources of PCBs, PCNs, and PCDD/Fs in the environment have been preliminarily screened by correlation analysis [11]. The similarity of the congener/homolog profiles of PCBs, PCNs, and PCDD/Fs between the samples and the known sources could be obtained from analysis results to judge the possible source. However, correlation analysis may not be able to screen all contamination sources because the target compounds in the sample are likely to come from varied sources with different congener patterns. In this case, positive matrix factorization (PMF) shows its advantages. A certain number of end member sources (factors) could be obtained by applying PFM based on a matrix of contaminant data. Then, the possible sources could be identified by comparing contaminant patterns between the factors and the known sources. Therefore, correlation analysis can be used to rapidly screen the possible sources in industrial parks, and then PMF can be applied to further analyze and verify these sources.

In this study, we focused on an industrial park located in Northwest China, which has a large electrolytic manganese production base with an annual production capacity of 500,000 tons. It is well known that manganese is indispensable in steel production. In recent years, increasing domestic demand for stainless steel and special steel, and increasing market demand for electrolytic manganese in China, have promoted the rapid development of the electrolytic manganese industry. Because of the extreme electricity consumption of electrolytic manganese and the need for timely and harmless treatment of hazardous wastes, such as electrolytic manganese slag, a power plant and a cement kiln have also been built in the industrial park. In addition, a steel smelter has been built nearby to facilitate the production of stainless steel and special steel. Although dioxins are not formed in the electrolytic manganese process, the industrial thermal processes in steel smelting, cement kiln processing, and power plants are potential sources of dioxins. It is worth noting that there is a large area of farmland around the industrial park. Because these factories are concentrated in one industrial park, contamination of the surrounding environment with dioxins should be evaluated.

To evaluate the impact of the industrial park on the surrounding environment, 13 soil samples were collected around the industrial park at sites selected according to the prevailing wind and potential distribution of dioxins from the sources. The concentrations of PCBs, PCNs, and PCDD/Fs were determined in the soil samples. After identifying fingerprints for the potential sources based on the literature, the sources of PCBs, PCNs, and PCDD/Fs were preliminarily determined using the congener profiles of these pollutants. Then, source apportionment of PCBs, PCNs and PCCD/Fs was conducted by a positive matrix factorization model based on the concentrations of all target congeners together. Finally, the health risk of human exposure to the three types of pollutants in the surrounding soil was assessed. The results will contribute to the prevention and treatment of environmental pollution with these toxic and harmful compounds.

## 2. Materials and Methods

### 2.1. Sample Collection

The industrial park is located on the north bank of the Yellow River in Ningxia Province. There are mountains behind the park, and the prevailing wind is from the northwest to the southeast. The industrial park was established in 2003 and covers an area of approximately 40 km^2^. Because of the ready availability of electricity, land resources, and a dry climate suitable for electrolytic manganese production, a large electrolytic manganese production base was built in the industrial park. As mentioned above, an iron and steel smelter, a cement kiln, and a power plant were also built in the industrial park. Because the local area is rich in agricultural straw as a resource, it is used as fuel for power plants in the industrial park. In addition, chemical fertilizer plants, agricultural product processing plants, sulfuric acid plants, and new material manufacturing plants have also been built in the industrial park. To investigate the concentrations of PCBs, PCNs, and PCDD/Fs in the soil around the industrial park, 13 soil samples (P1–P13) were collected in June 2017 (Figure 1). For each sample, surface soil (0–10 cm depth) was collected with a clean stainless steel shovel, placed in a clean opaque bag, and the bag was then sealed. After collection, samples were transferred to the laboratory as soon as possible and stored in a refrigerator at −18 °C.

Sample P4 was collected near the iron and steel smelter, and samples P5, P6, P11, and P12 were collected downwind of the smelter. Samples P7 to P10 were collected downwind from the cement kiln, and close to the iron and steel smelter and biomass power plant. Samples P1, P2, and P3 were collected upwind of the iron and steel smelter, and distant from the biomass power plant and cement kiln. Samples P1, P2, and P8 were collected from the landscaped green area in the industrial park. The land used in the industrial park was mainly obtained by pushing mounds to fill gullies, so the soil of the landscaped green area in the industrial park was derived from the new soil excavated during the construction of the industrial park. All other samples were collected from the surrounding farmland. Corn was the main crop planted in the local area. A background soil sample (P14) was collected from farmland located 25 km away from the industrial park, where corn was also the main crop. Moreover, it was distant from any potential dioxin sources.

### 2.2. Sample Analysis

The soil samples were freeze-dried before analysis. Known masses of ^13^C_12_-labelled standards of 17 2,3,7,8-chlorosubstituted PCDD/Fs, 12 dioxin-like PCBs (dl-PCBs), 4 indicator PCBs (id-PCBs), and ^13^C_10_-labelled standards of 6 PCNs were added to each soil sample (20 g). The samples were then extracted by accelerated solvent extraction with hexane/dichloromethane (50:50, *v/v*). The extraction temperature was 120 °C, and the extraction pressure was 1500 psi. Each extract was concentrated to 1–2 mL on a rotary evaporator, and then purified with an acidic silica gel column (from bottom to top: 1.0 g of activated silica gel, 8.0 g of silica/H_2_SO_4_ 44% (*w/w*) gel, 1.0 g of activated silica gel, and 4.0 g of anhydrous sodium sulfate) and a composite silica gel column (from bottom to top: 1.0 g of activated silica gel, 2.0 g of silica/AgNO_3_ 10% (*w/w*) gel, 1.0 g of activated silica gel, 5.0 g of silica/NaOH (1 M) 33% (*w/w*) gel, 1.0 g of activated silica gel, 8.0 g of silica/H_2_SO_4_ 44% (*w/w*) gel, 1.0 g of activated silica gel, and 4.0 g of anhydrous sodium sulfate). In the purification process with silica gel columns, 70 mL of hexane was used for pre-elution and 90 mL of hexane was used for elution. The PCCD/Fs were separated from the PCNs and PCBs using an alkaline alumina column (from bottom to top: 8.0 g alkaline alumina and 4.0 ganhydrous sodium sulfate). In this process, the PCNs and PCBs were first obtained by eluting the column with 100 mL of hexane/dichloromethane (95:5, *v/v*). The PCDD/Fs were then eluted with 50 mL of hexane/dichloromethane (50:50, *v/v*). The eluents were concentrated to 50 μL under a gentle stream of nitrogen and the concentrations of PCBs, PCNs, and PCDD/Fs were determined by gas chromatography tandem triple quadrupole mass spectrometry (Trace 1310-TSQ 8000, Thermo Fisher Scientific, Waltham, MA, USA). A DB-5 MS column (60 m × 0.25 mm i.d., 0.25 μm film thickness; J & W Scientific, Folsom, CA, USA) was used to separate the target compounds. The carrier gas was high-purity helium with a flow rate of 1.0 mL/min. Electron ionization was performed using an ion source operated at 270 °C and 70 eV. Multiple reaction monitoring mode was used for quantitative analysis. Other specific parameters are detailed in a previous paper [12].

### 2.3. Quality Assurance and Quality Control

The contents of target compounds in the soil samples were determined by the isotope dilution method. The detection limits for PCBs, PCNs, and PCDD/Fs were 0.11–0.79, 0.03–0.20, and 0.24–0.61 pg/g, respectively. The recovery ranges for ^13^C-labelled PCBs, PCNs, and PCDD/Fs were 49–80%, 51–82%, and 47–104%, respectively, which met the requirements for the determination of trace organic pollutants in environmental media [13,14]. A blank sample was analyzed for each batch of six to seven soil samples and treated in the same way as the soil samples. In the blank samples, some PCB and PCN congeners with low levels of chlorination were detected, but their contents were all less than 5% of those in the samples. Therefore, the soil samples were not blank corrected.

### 2.4. Data Analysis

Correlation analysis was performed using SPSS 13.0 for Windows. The positive matrix factorization model (PMF 5.0) was used to identify possible sources of PCBs, PCNs, and PCDD/Fs in the soil around the industrial park [15]. The congener concentrations of PCBs, PCNs, and PCDD/Fs in 13 soil samples were treated as the input data matrix. Detailed information on PMF can be found in the Appendix A, such as the determination of the number of factors and some of the outputs from the PMF. The carcinogenic risk (CR) and non-carcinogenic risk (no-CR) associated with exposure to PCBs, PCNs, and PCDD/Fs in soil was assessed using the model in the risk-assessment guidelines for the US EPA Superfund [16,17]. The calculation formulas and relevant parameters can be found in the Appendix A.

## 3. Results and Discussion

### 3.1. PCB, PCN, and PCDD/F Concentrations

The concentrations of target compounds in the soil samples are calculated by dry weight and the concentration of each congener can be found in Appendix A. The concentration ranges of dl-PCBs, id-PCBs and CB-209 in the soil samples from around the industrial area were 1.99–29.9, 9.76–151, and not detected (n.d.) –1120 pg/g (average 10.7, 80.7, and 450 pg/g), respectively. The total concentration (Σ_75_PCNs) range of 75 PCN congeners was 141–832 pg/g (average 596 pg/g). The total concentration (ΣPCDD/Fs) range of seventeen 2,3,7,8-PCDD/Fs congeners was 3.60–156 pg/g (average 53.5 pg/g). The concentrations of dl-PCBs, indicator, and CB-209 in the background soil sample were 2.5, 70.4, and 99.3 pg/g, respectively. The Σ_75_PCNs and ΣPCDD/Fs concentrations in the background soil sample were 11.4 and 6.92 pg/g, respectively. The World Health Organization 2005 toxic equivalency (TEQ) factor was used to calculate the TEQs of the PCDD/Fs and PCBs [18]. The toxicity conversion factors of PCN congeners relative to 2,3,7,8-TCDD summarized by Noma et al. [19] were used to calculate the TEQs of the PCNs. The TEQ concentration ranges and averages of the soil samples around the industrial area were 0.004–0.27 pg/g (average 0.05 pg/g) for the PCBs, 0.02–0.38 pg/g (average 0.11 pg/g) for the PCNs, and 0.14–1.51 pg/g (average 0.56 pg/g) for the PCDD/Fs. The TEQ concentrations of PCBs, PCNs, and PCDD/Fs in the background soil sample were 0.02 pg/g, 0.0007 pg/g, and 0.07 pg/g, respectively. Appendix A shows the percentages of PCBs, PCNs, PCDDs, and PCDFs by mass and TEQ in soil samples collected around the industrial park. PCNs were the predominant contributor (75–89%) to the total POPs (PCBs + PCNs + PCDDs + PCDFs) in soil samples P1, P2, and P8, which were collected from the landscaped green area in the industrial park. However, PCBs were the main contributor (29–70%) in other soil samples collected from farmland. It is worth noting that the high content of CB-209 in the soil samples collected from farmland was the reason PCBs accounted for a relatively high proportion of the total POP concentrations. The contribution of CB-209 to the total POP concentration ranged from 20% to 67%, while the contributions of dl-PCBs and id-PCBs were only 0.40–1.3% and 2.6–12%, respectively. Although the contribution of PCBs and PCNs to the total POPs concentrations was as high as 87–98%, PCDD/Fs had the largest contribution to the total TEQ concentration of the POPs (61–89%).

The spatial distribution of PCB, PCN, and PCDD/F concentrations in the soil around the industrial park is shown in Figure 1. The highest concentration of PCDD/Fs (156 pg/g) was observed in sample P4, which was collected closer to the iron and steel smelter than the other samples. The highest PCB concentration (1240 pg/g) was detected in sample P9, which was collected from farmland, while the lowest concentration (13.2 pg/g) was detected in sample P1, which was collected from the landscaped green area in the industrial park (i.e., not farmland). In addition, high concentrations of PCNs (638–832 pg/g) were found in samples P4 and P6–10. This result indicated that the sources of PCBs, PCNs, and PCDD/Fs in soil around the industrial park might be different. Interestingly, relatively high concentrations of CB-209 (n.d.–1120 pg/g), CN-75 (3.16–227 pg/g) and OCDF (1.19–118 pg/g) were observed in soil samples. Furthermore, the concentrations of CB-209 (195–1120 pg/g) and OCDF (11.3–118 pg/g) in samples collected from surrounding farmland were much higher than those from the landscaped green area (P1, P2 and P8) in the industrial park. This suggested that there might be a specific source of CB-209 and OCDF in the farmland soil.

To obtain a better understanding of soil contamination levels with PCBs, PCNs, and PCDD/Fs around the industrial park, the concentrations of these compounds in other domestic and international studies were collected and compared with this study (Table 1). The concentrations of PCBs in the soil around the industrial park (dl-PCBs: 1.99–29.9 pg/g, id-PCBs: 9.76–151 pg/g, and CB-209: n.d.–1120 pg/g) were comparable to those in soil around industrial thermal processes (dl-PCBs: 0.14–14 pg/g, and id-PCBs: 1.2–211 pg/g) in Slovakia, and around a municipal solid waste incinerator (dl-PCBs: 4.97–43.2 pg/g, and id-PCBs: 24.8–241 pg/g) in North China [20], but were lower than those in soil around industrial areas in North China (dl-PCBs: 13.9–229.1 pg/g) [21], in Dilovasi (average for id-PCBs: 39600 pg/g) and Iskenderun (average for id-PCBs: 7100 pg/g) in Turkey [22,23], and in Riyadh (dl-PCBs: 44–691 pg/g, and id-PCBs: 116–4219 pg/g) and Dammam (dl-PCBs: 38–360 pg/g, and id-PCBs: 104–871 pg/g) in Saudi Arabia [24]. Meanwhile, the concentrations of id-PCBs were higher than those of dl-PCBs in soil samples in these previous studies, which was consistent with the results of this study. Only half of the studies listed in Table 1 reported the CB-209 concentration (n.d.–100 pg/g) in the soil around the industrial area, which was lower than that in this study (n.d.–1120 pg/g). For PCNs, the concentrations of this study (Σ_32_PCNs: 51.9–520 pg/g, and Σ_4-8_PCNs: 23.7–472 pg/g) were similar to the total concentration of 32 PCN congeners in soil from an industrial area in Turkey (Σ_32_PCNs: 40–940 pg/g) and the total concentration of tetra- to octa-chlorinated naphthalenes from an industrialized chemical/petrochemical area in Spain (Σ_4-8_PCNs: 0–371.5 pg/g) [23,25], but were lower than those in soil around scrap processing steel plants in Turkey (Σ_32_PCNs: 3–10020 pg/g) [26], and a municipal solid waste incinerator (Σ_75_PCNs: 890–5410 pg/g) [27] and an industrial area (Σ_75_PCNs: 2194.4 pg/g) in North China [21]. In addition, the concentrations of CN-75 (n.d.–300 pg/g) were also determined in most of the previous studies listed in Table 1, which contributed little to the total concentration of PCNs. However, in this study, the concentrations of CN-75 (3.16–227 pg/g) contributed to nearly one tenth of Σ_75_PCNs (141–832 pg/g) on average, and were comparable to or higher than other studies. With regrowing to PCDD/Fs, the concentrations in the soil around the industrial park were similar to those in soil around two municipal solid waste incinerators (average 41.4 pg/g) [28] in Northeast China, and an iron and steel smelter (13–320 pg/g) [29] and an industrial area (average 101.8 pg/g) [21] in North China. However, the concentrations were lower than those in soil around a secondary aluminum smelter (23–3104 pg/g) [30] in Italy and an industrial area (1320 pg/g) [31] in the Pearl River Delta, China. At the same time, the concentrations of OCDD in the soil of the previous studies listed in Table 1 were all higher than those of OCDF, but the OCDF concentration in the present study was obviously higher than that of OCDD. Furthermore, the OCDF concentration in this study was comparable to or higher than that in other studies. In summary, although the contamination levels of PCBs, PCNs, and PCDD/Fs in soil around the industrial park in the present study are not higher than those in industrial areas in other regions, soil contamination levels with CB-209, CN-75, and OCDF are generally comparable to or higher than other studies and deserve our attention.

### 3.2. PCB, PCN, and PCDD/F Patterns

#### 3.2.1. PCBs

The PCB congener patterns in the soil samples around the industrial park are shown in Figure 2a. CB-209 was the predominant PCB congener in most of the soil samples, except for P1, P2, and P8, and accounted for 61–96% of the total concentration of 19 PCBs (ΣPCBs). CB-209 contributed more to the ΣPCBs in these soil samples than dl-PCBs (0.64–3.6%) or id-PCBs (3.7–36%). In addition, as mentioned in Section 2.1, all soil samples were collected from farmland except P1, P2, and P8. Because of the contribution of CB-209, the concentration of PCBs in the farmland soil samples was significantly higher than that in the soil collected from the landscaped green area in the industrial park (P1, P2, and P8). Therefore, PCB contamination was mainly affected by CB-209 in farmland soil around the industrial park.

As mentioned above, CB-209 was not detected or its concentration was low in the soil around the industrial area in the previous study (Table 1). Nevertheless, Tremolada et al. [36] found that CB-209 unexpectedly showed high concentrations (31–364 pg/g) with respect to the other congeners in soil from a high-altitude mountain pasture in the Italian Alps. On the other hand, some other studies have reported high proportions of CB-209 in water and sediment samples. Howell et al. [37] found that CB-209 accounted for a relatively high proportion of all PCBs in water and sediment samples from the Houston Shipping Channel. Huo et al. [38] reported that CB-209 was the main congener (45.5–83.9% of the total PCB concentration) in sediment from Chao Lake, China. Hartmann et al. [39] found that CB-209 accounted for a relatively high proportion of all PCBs in surface sediments from Narragansett Bay. CB-209 was detected as a major PCB congener in the water and sediment from the Delaware River [40,41]. In addition, Hermanson et al. [42] found that CB-209 was the most abundant congener in 10 of 27 tree samples collected near former manufacturing and incineration facilities in Sauget, Illinois, United States. Several potential sources of CB-209 in the environment were suggested in these studies, including inadvertent formation during the carbochlorination process of purifying titanium dioxide, impurity in phthalocyanine-type paint pigments, and legacy production of PCB mixtures containing CB-209 [36,38,40,41,42,43].

**Table 1 ijerph-20-03478-t001:** The concentrations (range or average) of PCBs, PCNs, and PCDD/Fs in soil samples collected around the industrial park (pg/g).

Compound	Countries and Regions	Industrial Thermal Processes		Concentration		TEQ	Reference
			dl-PCBs ^a^	id-PCBs ^b^	CB-209		
PCBs	North China	municipal solid waste incinerator	4.97–43.2	24.8–241	/ ^c^	0.02–0.18	[20]
	North China	comprehensive industrial area	13.9–229	/ ^c^	/ ^c^	0.12–0.94	[21]
	Dilovasi, Turkey	heavily industrial area	/ ^c^	39600	100	0.012–10.2	[22]
	Hatay-Iskenderun, Turkey	iron–steel plant	/ ^c^	7100	n.d. ^d^	/ ^c^	[23]
	Oissel, France	incinerator	/ ^c^	50.3	0.44	/ ^c^	[44]
	Notre-Dame de Gravenchon, France	refineries, chemical industries, and an incinerator	/ ^c^	150	n.d. ^d^	/ ^c^	[44]
	Slovakia	waste incinerator, metallurgical plants	0.14–14	1.2–211	/ ^c^	0.069–6.3	[45]
	Riyadh, Saudi Arabia	cement kiln, oil refinery, electric power plant, and steel industry	44–691	116–4219	/ ^c^	0.34–1.97	[24]
	Dammam, Saudi Arabia	cement kiln, oil refinery, steel industry, and desalination plant	38–360	104–871	/ ^c^	0.34–1.06	[24]
	Northwest China	comprehensive industrial area	1.99–29.9	9.76–151	n.d.–1120	0.004–0.27	This study
			Σ_75_PCNs ^e^	Σ_32_PCNs ^f^	CN-75		
PCNs	North China	municipal solid waste incinerator	890–5410	/ ^c^	/ ^c^	0.008–0.13	[27]
	North China	comprehensive industrial area	2194.4	283.2	n.d. ^d^ –104.8	0.02–0.92	[21]
	Aliaga, Turkey	electric-arc furnaces	/ ^c^	700	5.9	/ ^c^	[26]
	Tarragona, Spain	chemical/petrochemical area	0–371.5 ^g^	/ ^c^	n.d. ^d^ –19.8	/ ^c^	[25]
	Hatay-Iskenderun, Turkey	iron–steel plant	/ ^c^	40–940	5	/ ^c^	[23]
	Dilovasi, Turkey	heavily industrial area	/ ^c^	40–7070	2–300	0.0001–1.48	[22]
	Northwest China	comprehensive industrial area	141–832	51.9–520	3.16–227	0.02–0.38	This study
			PCDD/Fs	OCDD	OCDF		
PCDD/Fs	Northeast China	municipal solid waste incinerator	41.4	12.5	7.02	2.00	[28]
	North China	comprehensive industrial area	101.8	28.5	19.3	1.53–17.19	[21]
	North China	iron–steel plant	13–320	4.0–120	0.1–6.3	0.16–4.5	[29]
	Pearl River Delta, China	comprehensive industrial area	1320	983	5.93	4.80	[31]
	Slovakia	waste incinerator, metallurgical plants	20–1027	28.02	1.46	0.28–15.9	[45]
	Piemonte, Italy	secondary aluminum smelter	23–3104	12.6–145	1.56–113	0.2–64.0	[30]
	Northwest China	comprehensive industrial area	3.60–156	0.56–34.4	1.19–118	0.14–1.51	This study

^a^ dioxin-like PCBs (12 congeners), ^b^ indicator PCBs (7 congeners), ^c^ not reported, ^d^ not detected, ^e^ mono- to octa-chloronaphthalenes (75 congeners), ^f^ tri- to octa-chloronaphthalenes (32 congeners), ^g^ tetra- to octa-chloronaphthalenes (number of congeners not reported).

The patterns of PCBs in various potential sources (especially those with high CB-209 concentrations) were collected to analyze the sources of PCBs in soils in this study. Unfortunately, to the best of our knowledge, data remains scarce concerning the inadvertent formation of PCBs from titanium dioxide purification. Nevertheless, a few studies have determined the concentrations of PCBs in white titanium pigment, the main component of which is titanium dioxide. Ctistis et al. [34] reported the concentrations of dl-PCBs and id-PCBs in titanium pigment in the Netherlands, but the concentrations of CB-209 were not reported. In the United States, CB-209 was not detected in white titanium pigment [7]. In the meantime, despite our best efforts, we failed to access the PCB congener profiles in commercial PCB formulations with a high CB-209 content, such as Aroclor 1268 and Aroclor 1270. Current knowledge indirectly indicates that there is a small amount of CB-209 (≤5%) in Aroclor 1268 [38], and CB-209 is the main component of Aroclor 1270 [42]. In contrast, many studies have reported the congener profiles of PCBs in commercial pigments [6,7,33,46]. High concentrations of CB-209 were detected in the phthalocyanine green pigments, and CB-209 was the predominant contributor to total PCBs [6,7]. In addition, PCBs could be unintentionally formed in industrial thermal processes, and trichlorinated PCBs (50–65%) were the main emitted congeners [35].

Profiles of 19 PCB congeners in soil samples in the present study (Figure 2a) are compared with those in commercial PCB formulations [32], phthalocyanine green pigments [6,33], a titanium pigment [34] and stack gas from steel smelting and cement kilns [35]. It was reported the pattern of main commercial PCB formulations from different countries, such as Aroclor, Kanechlor and Clophen, was similar [47]. Among them, Aroclors accounts for 50% of the total PCB products [48,49], so we choose it as the representative of commercial PCB formulations. CB-28 was the main congener in five commercial PCB formulations (Aroclor 1016, 1221, 1232, 1242, and 1248) and stack gas from steel smelting and cement kilns. CB-180, CB-138 and CB-153 were the main congeners in two commercial PCB formulations (Aroclor 1260 and Aroclor 1262) and the titanium pigment. CB-209 is the predominant congener in phthalocyanine green pigments, accounting for nearly 100% of the total 19 PCB congeners. In soil samples, CB-28 was the main congener in samples P1, P2 and P8, but CB-209 was the predominant in other samples. It seemed unlikely to identify the sources of PCBs in soil from these comparisons due to the diversity of PCB profiles in soil samples. However, the PCB congener profiles in soil samples were similar (r > 0.78, *p* < 0.01) to each other (Figure 2b) when CB-209 is removed. It is indicated that CB-209 in farmland soil may come from a specific source, which mainly affects the contamination of CB-209 and has little impact on other PCB congeners in farmland soil. In other words, it could be inferred that CB-209 was from a specific source, where CB-209 was the predominant congener and the proportion of other congeners was low or absent.

#### 3.2.2. PCNs

The PCN homolog patterns in the soil samples collected around the industrial park are shown in Figure 3. PCNs with lower levels of chlorination (mono- to trichloronaphthalenes) were the main homologs (54–89% of the Σ_75_PCNs), except in soil sample P4. In sample P4, octachloronaphthalene (CN-75) was the predominant homolog, accounting for 32% of the Σ_75_PCNs. In addition, the proportions of CN-75 were also relatively high in soil samples P1, P7, and P11, where it accounted for 14%, 28%, and 12% of the Σ_75_PCNs, respectively. The various patterns of PCNs homologs in soil samples indicated that there might be multiple sources of PCNs in the industrial park. It is generally believed that there are two main sources of PCNs in the environment: (1) legacy commercial PCN or PCB formulations, and (2) unintentional emissions from industrial thermal processes. The ratio of the total PCN combustion-related congener content to the total PCNs content (∑_com_PCNs/∑PCNs) is often used to preliminarily identify the source of PCNs in the environment. The ratios of ∑_com_PCNs/∑PCNs were in the range of 0.12–0.25 in the soil samples in the present study, which were greater than those of Halowax commercial PCN formulations (<0.11) but less than those of fly ashes from various combustion processes (0.5–0.75) [50]. This may imply that re-emissions of historically used commercial PCN formulations and combustion processes contribute together to PCNs in the soil around the industrial park.

There are usually specific PCN homolog patterns for emissions from various industrial sources and commercial formulations [53]. Therefore, the proportions of PCN homologs were obtained in stack gas samples from five converter steel plants (CS1–5) [51] and a cement kiln [52], seven commercial PCN formulations (Halowax 1000, 1001, 1013, 1014, 1031, 1051, and 1099) [19], and a commercial PCB formulation (PCB 3#) [47] from the literature, and comparisons were made with the proportions of PCN homologs in soil samples (Figure 3). Historically, the Halowax series has been the most commonly used among commercial PCN formulations. PCB 3# is a commercial PCB formulation that was widely used as insulating oil in transformers and capacitors in China, in which a relative high concentration of PCNs was detected by Huang et al. [47]. Mono- to trichloronaphthalenes were also the main homologs in the Halowax 1031, 1000, and stack gas from the converter steel plants and the cement kiln, and their homolog profiles were more similar to those in the soil samples (except for P4, P7 and P11, which had a high proportion of CN-75) based on the correlation analysis of homolog profiles (Appendix A). This result indicated that PCN contamination may be mainly affected by steel plants, cement kilns and the legacy of commercial PCN formulations (Halowax 1031 and Halowax 1000) in the industrial park.

#### 3.2.3. PCDD/Fs

Figure 4 shows the PCDD/F congener profiles in soil samples collected around the industrial park. In most of the 13 samples (except P1, P2, and P8), OCDF was the predominant congener, accounting for 62–89% of the ΣPCDD/Fs. In soil samples P1 and P2, the main congeners were OCDF (25.4% and 33.0%), 1,2,3,4,6,7,8-HpCDF (19.5% and 18.6%), and OCDD (15.8% and 15.6%). Unlike the other samples, OCDD (58.4%) was the predominant congener in P8, followed by OCDF (20.3%). Interestingly, OCDF was observed to be the most dominant congener in most of soil samples (except P8) in the present study, while OCDD was frequently determined to be the most dominant congener in environmental soil samples [29,30,54]. Therefore, a specific source of PCDD/Fs in soil might exist around the industrial park. Industrial thermal processes have been reported to be the main source of PCDD/Fs in the environment in recent years. In addition, the specific profiles of PCDD/Fs were observed in chemical products, such as chlorophenols and chlorophenoxy acid herbicides [4,5]. Therefore, PCDD/F congener profiles were drawn in stack gas emitted from industrial thermal processes (including converter steel plants, cement kilns, secondary copper smelters, secondary aluminum smelters, secondary lead smelters, and waste incinerators) [51,52,55] and chemical products containing PCDD/Fs (including a pentachlorophenol, a sodium pentachlorophenate, and 2,4-D products) [4,5], and the profiles were compared with those in soil samples (Figure 4). The proportion of OCDF in the potential sources mentioned above is comparable to or lower than those of OCDD except for a 2,4-D butyl ester (D5) produced in China. Furthermore, PCDD/F congener profiles in D5 were similar to those in the soil samples (except for P8, which had a high proportion of OCDD) based on the correlation analysis of congener profiles (r > 0.88, *p* < 0.01). The 2,4-D butyl ester is an auxinic herbicide widely used in agriculture, and that can effectively control weeds [5,56]. Perhaps similar 2,4-D products have been used in this study area. Fortunately, as the registration certificate of the last 2,4-D product available for domestic use has expired and would not be renewed on 28 January 2021, the production of 2,4-D products for domestic supply has been completely stopped in China since then [57]. It is likely that the contamination of PCDD/Fs in the soil around the industrial park may be mainly caused by the historical use of 2,4-D products.

### 3.3. Positive Matrix Factorization (PMF)

From the above analysis, it can be inferred that there may be multiple sources of PCBs, PCNs, and PCDD/Fs in the soil around the industrial park. Therefore, the PMF model was further applied to identify the potential sources of PCBs, PCNs, and PCDD/Fs in soil around the industrial park. PCBs, PCNs, and PCDD/Fs in the environment often have the same source, such as industrial thermal processes. Furthermore, some commercial PCB and PCN formulations and chemicals also simultaneously contain these three kinds of compounds [19,47,58]. Therefore, we simultaneously input the concentrations of all PCB, PCN, and PCDD/F congeners in each soil sample as the original data for analysis when using PMF for source apportionment. The four resolved source profiles (factors) generated by the PMF model are shown in Figure 5. The reasons for selecting four factors are detailed in Appendix A.

Factor 1 accounted for 16.6% of total POPs (PCBs + PCNs + PCDD/Fs) in the data set. In factor 1, OCDF (7.30%), CB-209 (35.4%), and CN-75 (19.5%) with the highest degree of chlorination were the main congeners. In some commercial formulations or chemical products, the highest chlorinated congeners of PCBs, PCNs, and PCDD/Fs occurred together. In a 2, 4-D product, OCDF and CB-209 were the main contributors of PCDD/Fs and PCBs, respectively [5]. CN-75 and OCDF were the predominant congeners of PCN and PCDD/F in octachlorinated commercial PCN formulations (Halowax 1051), respectively [19]. Furthermore, the segments of PCNs and PCDD/Fs in factor 1 resembled Halowax 1051 in PCN congener profiles (*r =* 0.954, *p <* 0.01) and the 2,4-D product in PCDD/F congener profiles (*r =* 0.865, *p <* 0.01). Therefore, factor 1 appeared to represent the legacy of commercial formulations or chemical products containing highly chlorinated PCBs, PCNs, and PCDD/Fs, such as Halowax 1051 and some 2,4-D products.

Factor 2 accounted for 27.9% of total POPs in the data set. Notably, CB-209 was the only predominant contributor to factor 2, accounting for 71.3% of the total POP concentrations. This was consistent with the speculation made in Section 3.2.1: there may be a specific contamination source, where CB-209 was the predominant congener and the proportion of other congeners was low or absent. Trichlorobiphenyl and pentachlorobiphenyl are the most widely used commercial PCB formulations in China [59], in which the contents of CB-209 are very low [32]. Meanwhile, emissions of CB-209 from industrial thermal processes were minimal [35]. Therefore, it was unlikely that the CB-209 in the soil samples collected in this study was mainly derived from industrial thermal processes or commercial PCB formulations. In previous studies, high concentrations of CB-209 were often detected in water and sediments in rivers and lakes [38,40,41,60], indicating that CB-209 can easily migrate to rivers or lakes. What is more, the historical deposition of CB-209 in Lake Ontario sediment was observed to be associated with pigment and dye production trends in the US, and the presence of CB-209 in sediment layers deposited before the period of the production of commercial PCB formulations [60]. In the present study, it was noteworthy that the soil samples (including background soil) with high CB-209 concentrations were all collected from farmland, while the concentrations of CB-209 in soil samples collected from the landscaped green area in the industrial park were relatively low or even not detected. This result indicated that the contamination of CB-209 might be related to farming activities. Because of the lack of rain in Northwest China, the farmland in the study area has been irrigated with water from the Yellow River by a canal system. An industrial park, where many plants of pigments and cashmere fabric printing and dyeing were located, was also located on the North Bank of the Yellow River, approximately 35 km west of the present industrial park, which was the upstream area of the current study area. Therefore, industrial wastewater rich in CB-209 discharge from these factories in the upstream area might have been transported to the farmland around the present industrial park and background soil point via the Yellow River and irrigation canals, and CB-209 from the wastewater could have accumulated in the soil after the water was used for irrigation. Finally, we speculated that factor 2 represented phthalocyanine pigments with high concentrations of CB-209.

Compared with factor 1 and factor 2, the kinds of three POP congeners were more abundant in factor 3 and factor 4. Factor 3 accounted for 27.5% of the total POPs in the data set and was dominated by mono- to tri-chlorinated naphthalenes and id-PCBs, which were also the main congeners of PCNs and PCBs emitted from industrial thermal processes [35,53]. Meanwhile, many highly chlorinated PCN combustion-related congeners were obviously observed in factor 3, such as CN-25/60, CN-50, CN-51, CN-54 and CN-66/67. Furthermore, the segments of PCNs, PCBs and PCDD/Fs in factor 3 resembled the homolog profiles of PCNs (*r =* 0.762, *p <* 0.05), and the congener profiles of PCBs (*r =* 0.895, *p <* 0.01) and PCDD/Fs (*r =* 0.860, *p <* 0.01) emitted from steel smelting, respectively [35,51]. Thus, factor 3 appeared to represent the POP emissions from the iron and steel smelters in the industrial park.

Factor 4 accounted for 28.1% of the total POPs in the data set and dominated CN24/14 (15.5%) and CN-5/7 (15.5%). CN-24 was found to be the main PCNs congener formed and emitted from coal and wood combustion [61]. Coal is the main fuel in the process of cement kiln production, and agricultural straw is the main fuel of the biomass power plant in the industrial park. CN-5/7 was the predominant congener in Halowax 1000 [19], but it was present in very low concentrations in stack gas emitted from industrial thermal processes based on a previous investigation [52,55]. In addition, factor 2 was dominated by low-chlorinated PCNs that have higher vapor pressures than other highly chlorinated homologues and easily tend to undergo long-range transport in the atmosphere. Thus, we hypothesized that factor 4 represented POP emissions from the cement kiln and the biomass power plant in the industrial park, as well as other sources, such as Halowax 1000 and atmospheric deposition of nonlocal sources.

According to the results of PMF, the contamination of CB-209, CN-75, and OCDF may be related to the legacy of Halowax 1051, as well as impurities in some chemicals, such as 2,4-D products with a high content of OCDFs and phthalocyanine pigments with a high content of CB-209, which together contributed 44.5% of the total concentration of POPs. The local industrial thermal processes were mainly responsible for the contamination of three POPs in the surrounding soil except for highly chlorinated POPs.

### 3.4. Health Risk Assessment

There are three main routes for human body exposure to pollutants in the soil, including ingestion, dermal contact and inhalation. The range of total no-CR associated with exposure to PCBs, PCNs, and PCDD/Fs in soil around the industrial park was 0.0039–0.037, which was far below the no-CR threshold (1.0). The range of total CR was 0.042 × 10^−6^–0.40 × 10^−6^, which was also below the CR threshold (1.0 × 10^−6^) (Appendix A). Wu et al. assessed the health risks of PCBs, PCNs, and PCDD/Fs in the soil of an industrial park in Eastern China, and found that a soil sample from near a secondary copper smelter had a high carcinogenic risk (0.85 × 10^−6^), but these pollutants exhibited low carcinogenic risk in all other samples [21]. Xu et al. evaluated PCNs, PCDD/Fs, and PCBs in soil samples collected in an industrial park in northwestern China, and found that there were four sites in the park with a carcinogenic risk (0.487 × 10^−6^, 0.234 × 10^−6^, 0.230 × 10^−6^ and 0.210 × 10^−6^) close to the risk threshold [43]. Al-Wabel et al. evaluated the pollution level, source and human health risk of PCBs in the soil of industrial areas in central and eastern Saudi Arabia. The adverse effects of PCBs on human health are very low (CR value ≤ 10^−6^) [24]. In general, the carcinogenic risk of electrolytic manganese industrial park in this study is slightly lower than that reported by Wu et al., higher than that reported by Al-Wabel et al., and close to that reported by Xu et al. [21,24,43].

For exposure routes, ingestion contributed highest (89.1% and 93.3%) to the total CR and no-CR, followed by dermal contact (7.5% and 6.7%) and inhalation (3.4% and 0.06%). With regard to compounds, PCDFs contributed highest (59.6%) to the total CR and no-CR, followed by PCNs (16.5%), PCDDs (15.8%) and PCBs (8.1%). These results are consistent with previous studies; ingestion is the predominant exposure route, and PCDD/Fs contribute highest to health risk [21,43]. It is worth noting that the CRs of some samples reached 0.40 × 10^−6^, 0.32 × 10^−6^, and 0.22 × 10^−6^ in this study. Since these pollutants can accumulate in the soil with increased time, so the levels of PCBs, PCNs, and PCDD/Fs in the surrounding soil deserve continuous attention.

## 4. Conclusions

This study investigated soil contamination with PCBs, PCNs, and PCDD/Fs around an industrial park in Northwest China. To the best of our knowledge, this is the first study to perform source apportionment applying PMF based on the concentrations of all congeners of PCBs, PCNs and PCDD/Fs together. The levels of these compounds are not higher than those around industrial areas in other regions, but the concentrations of CB-209, CN-75, and OCDF are generally higher than others, and have their own unique contamination characteristics. The contamination of highly chlorinated congeners of PCBs, PCNs, and PCDD/Fs might be derived from phthalocyanine pigments with high content of CB-209, the legacy of Halowax 1051, and 2,4-D products with high content of OCDF, respectively. In addition, local industrial thermal processes were mainly responsible for the contamination of PCBs, PCNs, and PCDD/Fs in the surrounding soil except for CB-209, CN-75, and OCDF. The results of PMF indicated that the contribution of chemical products to soil contamination was nearly equivalent to that of industrial thermal processes. Therefore, the contamination of chemical products containing highly chlorinated POPs should attract far more attention. Although the exposure risks to PCBs, PCNs, and PCDD/Fs in soil samples are all lower than the reference threshold, the carcinogenic risks at several points deserve continuous attention.

## Figures and Tables

**Figure 1 ijerph-20-03478-f001:**
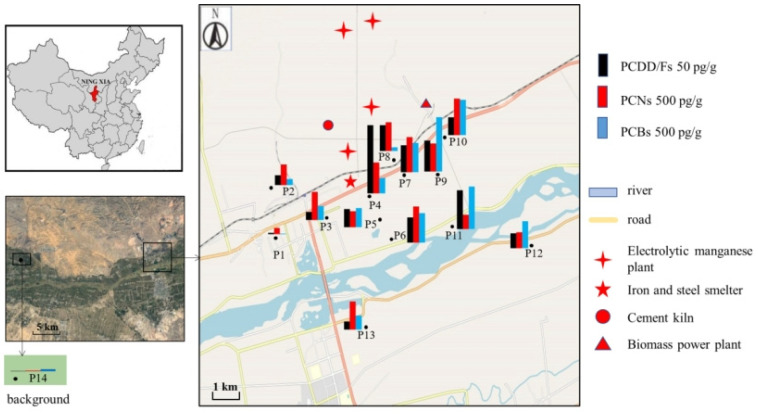
Soil sampling locations and concentrations of PCBs, PCDD/Fs, and PCNs in soil samples collected around the industrial park.

**Figure 2 ijerph-20-03478-f002:**
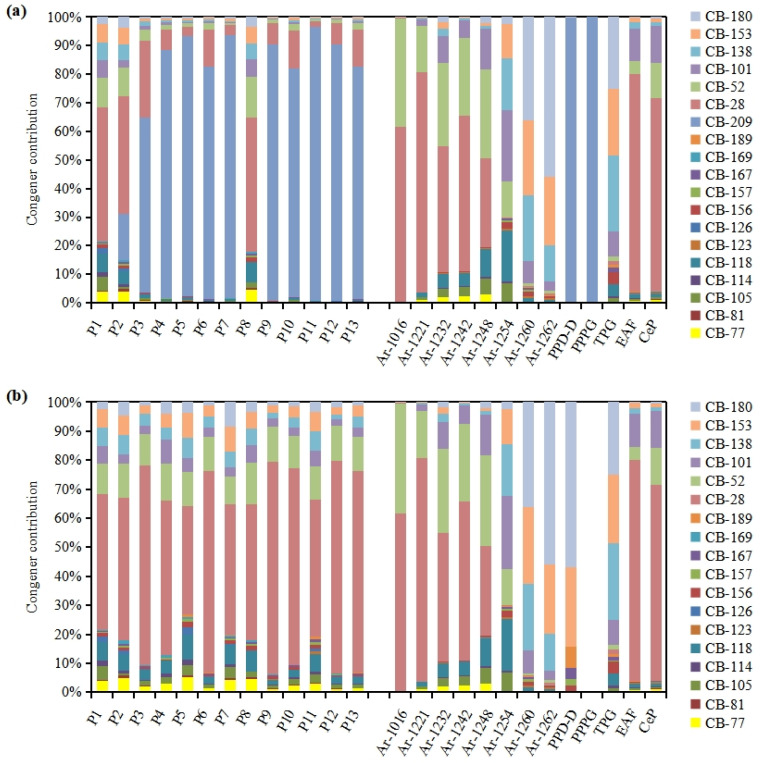
PCB patterrns with (**a**) and without CB-209 (**b**) in soil samples of the present study, commercial PCB formulations (Ar-1016, 1221, 1232, 1242, 1248, 1254, 1260, 1262) [32], phthalocyanine green pigments (PPD-D and PPPG) [6,33], a titanium pigment (TPG) [34] and stack gas from steel smelting (EAF) and cement kiln (CeP) [35] (only CB-209 were detected in PPPG according to the congeners reported).

**Figure 3 ijerph-20-03478-f003:**
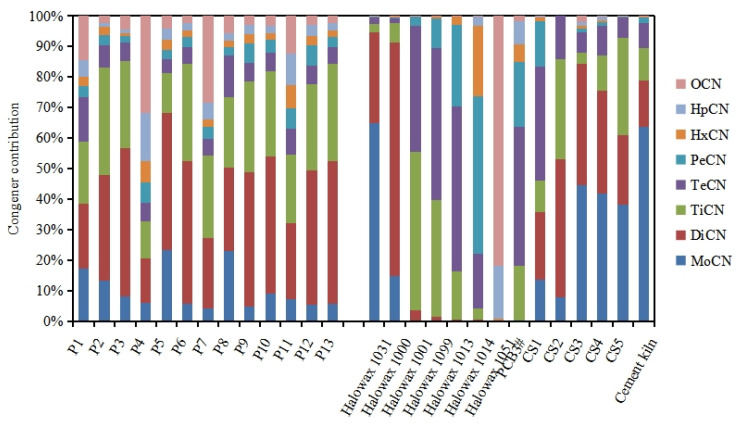
PCN patterns in the soil samples (P1–13), flue gas samples from five converter steel plants (CS1–5) [51] and a cement kiln [52], seven commercial PCN formulations (Halowax 1000, 1001, 1013, 1014, 1031, 1051, and 1099) [19], and a commercial PCB formulation (PCB 3#) [47].

**Figure 4 ijerph-20-03478-f004:**
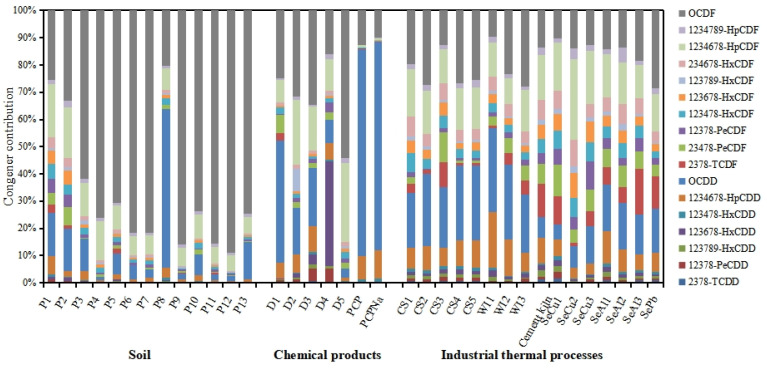
PCDD/F congener patterns in soil samples, 2,4-D butyl esters (D1–5) [5], a pentachlorophenol (PCP), a sodium pentachlorophenate (PCPNa) [4], and stack gas of converter steel plants (CS1–5) [51], cement kiln, secondary coper smelters (SeCu1–3), secondary aluminum smelters (SeAl1–3), secondary lead smelter (SePb) and waste incinerators (WI1–3) [52,55].

**Figure 5 ijerph-20-03478-f005:**
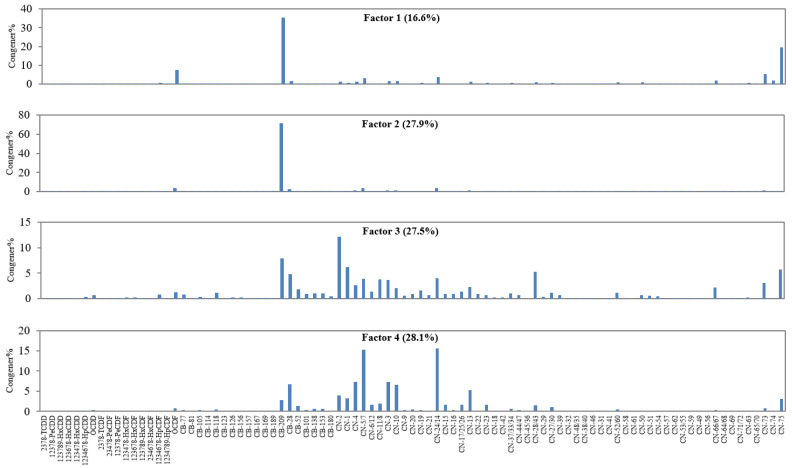
Congener profiles of four resolved sources (Factors) generated by PMF modle.

## Data Availability

The data presented in this study are available in Appendix A.

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
