# Peer review of "PCBs, PCNs, and PCDD/Fs in Soil around an Industrial Park in Northwest China: Levels, Source Apportionment, and Human Health Risk"

_ijerph, 2023, doi:10.3390/ijerph20043478_

Round 1

Reviewer 1 Report

The levels of the toxic organic pollutants in soil are vital for the assessment of the soil pollution, as well as the land planning made by the governments. In this work, the authors determined the concentrations of polychlorinated biphenyls (PCBs), polychlorinated naphthalenes (PCNs), and polychlorinated dibenzo-p-dioxins and dibenzofurans (PCDD/Fs) around an industrial park in Northwest China and also investigate the potential impacts of emissions of the park on the surrounding environment. However, the current ms still suffered from some issued that needed to be addressed clearly.

(1) The abstract is too long. The abstract should show the key results of the study, not the specific results. Please shorten and rewrite it.

(2) Language needs improvement. Please work through the ms carefully from this perspective.

(3) In the introduction, the significance of this work is unable to be understand, please well organize this section to clearly show the novelty of this work.

(4) The discussion of the contaminants on the health risk assessment is too limited, please supplement some deep discussion in this section based on the similar literatures.

(5) What is the significance of the data related to the levels of the organic pollutants? This work just showed the levels of these pollutants and their potential sources. Will the current pollutants greatly impact the human activities or the industrial production? Or do the soil is needed to be remediated immediately? 

Reviewer 2 Report

My humble suggestions for the authors are :

1.      Indicate the citation after the categorical statements. For instance Polychlorinated biphenyls (PCBs), polychlorinated naph-37 thalenes (PCNs), and polychlorinated dibenzo-p-dioxins and 38 dibenzofurans (PCDD/Fs) are listed in the Stockholm Conven-39 tion on persistent organic pollutants (POPs) because they are re-40 sistant to degradation, highly toxic, can be enriched through the 41 food chain, and can migrate globally via various pathways. Apart 42 from for scientific research, humans have never intentionally 43 produced PCDD/Fs. Ref. Stockholm.

2.      Mention the quality control or assurance in carried out.

3.      Show the USEPA formula for risk assessment.

Reviewer 3 Report

Comments on Manuscript ijerph-2018332 “PCBs, PCNs, and PCDD/Fs in soil around an industrial park in Northwest China: Levels, source apportionment, and human health risk”

This manuscript by Li et al., reports the soil concentrations of PCBs, PCNs and PCDD/Fs from an industrial park in Northwest China. The manuscript is well written and easy to follow. However, the only issue that I have with this manuscript is about the health risk assessment. I believe the authors do not have enough data to perform this assessment, especially the inhalation risk assessment (i.e., air samples). There are so many variables with unknown uncertainties that it makes this type of analysis meaningless. Just because the U.S. EPA has guidelines on how to perform a health risk assessment, it does not mean it will have a scientific value. Perhaps in a report will be fine, but not in a scientific peer reviewed journal, at least with the available data in this manuscript. I would suggest to the authors to eliminate at least the inhalation risk calculations. Further, there are missing figures in the version of the manuscript that I reviewed. That is, I could not find figures 2, 3, 4 and 5. Thus, I am not sure if I can accept this manuscript for publication without evaluating those figures. I have a few comments, mostly clarifications:

Line 74. I would suggest to the authors to use signature or profile instead of characteristics.

Sample Analysis section. It is not clear how many congeners were analyzed for PCBs, PCNs and PCDD/Fs. This information can be included in the SI. This would also help understand better the PMF results. Further, are the concentration values dry weight? They should be.

Did the authors think about measuring organic matter (i.e., organic carbon) in the soil? That could help understand better the spatial distribution of the investigated pollutants.

Data analysis section. I was wondering why the authors did not perform any type of spatial analysis or distance to potential source(s) regressions. Perhaps a negative result could also be interesting.

Finally, I would recommend the authors to add/upload the data to an online data repository. That will make the data more findable, accessible, interoperable, and reusable, i.e., FAIR. At the end, if someone would like to use it, it will be easy to find it.

Round 2

Reviewer 1 Report

The manuscript has been well revised based on the comments and meets the requirements of the journal.

Author Response

Thanks very much for taking your time to review this manuscript, and thank you for your affirmation of our research. Best regards to you.

Reviewer 3 Report

Comments on Manuscript ijerph-2018332 “PCBs, PCNs, and PCDD/Fs in soil around an industrial park in Northwest China: Levels, source apportionment, and human health risk”, Revision 2.

The authors addressed most of my questions and comments but also raised a few more questions with their answers.

With only 19 PCB congeners, which represents only 9.09% of all PCB congeners (19/209), it is not possible to realistically create a representative PCB profile. There is a significant amount of missing information, including important congeners such as PCBs 8, 11, and 31. The authors should consider this. However, if they had measured all the congeners in a few samples and observed that the total PCB concentration with only 19 congeners is not significantly different from the total PCBs, it might make sense to only use these 19 congeners.

The other issue that I have is regarding the use of PMF. This is the first time I have seen different POPs being gathered to perform this analysis, but what is more important is the explanation for why they did it. The authors’ statement is not entirely accurate:

The PCBs, PCNs, and PCDD/Fs in the environment often have the same source, such as industrial thermal processes (lines 426 – 427). That is possible, but the opposite is also possible and more probable. Most PCB contamination is likely to come from the use of hydraulic fluids, not thermal processes. Thus, it doesn’t make much sense to combine all the data to perform a PMF. Further, it is interesting to note that the authors are comparing the “reduce” PCB profiles of their samples against Aroclors, which were not widely used in China. This needs to be explained.

Finally, as I mentioned previously with the risk assessment, the authors are attempting to do too much with their limited amount of data. This greatly weakens the quality of the paper. The authors should carefully consider what they can realistically achieve with their data and not exceed those bounds.
